

# Links between Baltic Sea submarine terraces and groundwater sapping

Martin Jakobsson[1], Matt O'Regan[1], Carl-Magnus Mörth[1], Christian Stranne[1], Elizabeth Weidner[1,2], Jim Hansson[4], Richard Gyllencreutz[1], Christoph Humborg[3], Tina Elfwing[3], Alf Norkko[5,3], Joanna Norkko[5],
Björn Nilsson[6], and Arne Sjöström[6]

[1]Department of Geological Sciences, Stockholm University, and Bolin Centre for Climate Research, Stockholm, 10691, Sweden
[2]Center for Coastal and Ocean Mapping, University of New Hampshire, USA
[3]The Baltic Sea Centre, Stockholm University, Stockholm, 10691, Sweden
[4]The Maritime Museum, Stockholm, Sweden
[5]Tvärminne Zoological Station, University of Helsinki, Hanko, Finland
[6]Department of Archaeology and Ancient History, Lund University, Sweden

*Correspondence to*: Martin Jakobsson (martin.jakobsson@geo.su.se)

## Abstract

Submarine Groundwater Discharge (SGD) influences ocean chemistry, circulation, spreading of nutrients and pollutants, and shapes seafloor morphology. In the Baltic Sea, SGD was linked to the development of terraces and semi-circular depressions mapped in an area of the southern Stockholm Archipelago, Sweden, in the 1990s. We mapped additional parts of the Stockholm
Archipelago, areas in Blekinge, southern Sweden, and southern Finland using high-resolution multibeam sonars and sub-bottom profilers to investigate if the seafloor morphological features discovered in the 1990s are widespread and to further address the hypothesis linking SGD to their formation. Sediment coring and seafloor photography conducted with a Remote Operated Vehicle (ROV) and divers add additional information to the geophysical mapping results. We find that terraces, with general bathymetric expressions of about 1 m and lateral extents of sometimes >100 m, are widespread in the surveyed areas
of the Baltic Sea and are consistently formed in glacial clay. Semi-circular depressions, however, are only found in a limited part of a surveyed area east of the island Askö, southern Stockholm Archipelago. Our study supports the basic hypothesis of terrace formation initially proposed in the 1990s, i.e. groundwater flows through siltier permeable layers in glacial clay to discharge at the seafloor, leading to the formation of a sharp terrace when the clay layers above seepage zones are undermined enough to collapse. By linking the terraces to a specific geologic setting, our study further refines the formation hypothesis
and forms the foundation for a future assessment of SGD in the Baltic Sea that may use marine geological mapping as a starting point. We propose that SGD through the sub-marine seafloor terraces is most likely intermittent and linked to periods of higher groundwater levels, implying that to quantify the contribution of freshwater to the Baltic Sea through this mechanism, more complex hydrogeological studies are required.





# 1 Introduction

The influence of groundwater on seafloor morphology has been discussed for more than 80 years within the geoscientific community (Robb, 1990). Early examples include Stetson (1936), who suggested that the formation of submarine canyons in

the southern flank of Georges Bank off the north eastern U.S. coast were related to groundwater sapping aided by currents. Submarine canyons are today linked to a combination of geological processes including erosion by turbidity currents, slumping, and mass wasting (Harris and Whiteway, 2011; Shepard, 1981). However, groundwater as a shaping agent of seafloor morphology appears in several other marine geological settings (Robb, 1990). For example, mapped terraced walls and irregular courses of valleys in the continental slope off New Jersey have been interpreted to be formed by groundwater

sapping during periods of lower sea level (Robb, 1984). Other examples of seafloor morphological expressions related to groundwater are depressions formed around submarine fresh/brackish springs in the Mediterranean, where extensive nearshore carbonate formations caused the development of submarine karstic aquifers (Rousakis et al., 2014). Depressions, which are similar in appearance to pockmarks formed by gas seeps, have also been found to form where freshwater escapes through the seafloor (Whiticar and Werner, 1981; Khandriche and Werner, 1995; Virtasalo et al., 2019). Groundwater discharge into the

ocean is recognized as a widely occurring process and is commonly referred to as Submarine Groundwater Discharge (SGD) (Moore, 2010). SGD is estimated to contribute about 6-7% of the total hydrological discharges to the world oceans (Zektser, 2000), although different quantification methods have yielded varying results at specific localities (Prieto and Destouni, 2011). In the Mediterranean Sea, SGD has been shown to be a major source of nutrients (Rodellas et al., 2015) and the process has generally been raised as a potentially underestimated provider of chemical elements, including pollutants and nutrients, from

land to coastal waters (Destouni et al., 2008).

In Sweden, terraces and semi-circular depressions in the Baltic seafloor were mapped in the 1990s along some islands in the southern Stockholm Archipelago (Söderberg and Flodén, 1995) (Fig. 1). These features were interpreted to be formed by processes related to SGD (Söderberg and Flodén, 1995). The proposed mechanism was groundwater flowing through siltier

permeable layers in glacial clay, and where the clay outcropped at the seafloor, the escaping fresh water lead to the development of a terrace when the above layers of clay were undermined and collapsed. With respect to the semi-circular depressions, Söderberg and Flodén (1995) found morphological similarities with seafloor features mapped near Kiel in Eckernförde Bay, southwestern Baltic Sea (Khandriche and Werner, 1995), where SGD is documented to occur from geochemical analyses of seafloor sediment and the water column (Schlüter et al., 2004).






More recently, about 1 m high seafloor terraces, extending from few meters to >100 m in length, and semi-circular depressions 10-30 m wide and about 1 m deep were mapped using high-resolution multibeam echo sounder and sub-bottom profiler east of the island Askö in the southern Stockholm Archipelago (Jakobsson et al., 2016). Both the terraces and depressions closely fit the descriptions by Söderberg and Flodén (1995), suggesting that these kind of seafloor features may be widespread along
the Swedish coast. Here we show from extended high-resolution geophysical surveys in Stockholm Archipelago and in Blekinge, southern Sweden, as well as in southern Finland, that seafloor morphological features in the form of terraces and semi-circular depressions are widespread in the Baltic Sea. Sediment cores and bottom inspection with ROV and divers reveal that the terraces consistently form in near identical geological settings, permitting prediction of their occurrence if information on the seafloor geology is available. In order to further test the formation hypothesis involving groundwater sapping, we
performed geochemical analyses of a carbonate concretion found in the glacial clay unit of a submarine terrace. This study provides a framework for continued investigations involving *in situ* monitoring of potential groundwater sapping at selected terraces and semi-circular depressions along Baltic Sea coasts. The widespread occurrence of these features suggests that SGD in the Baltic Sea may be an important influence on the chemical composition of its waters with implications for circulation and the spread of nutrients and pollutants.

## 2 Material and Methods

### 2.1. Survey areas

Results are presented from four different regions in the Baltic Sea: (1) east of the island Askö in Southern Stockholm Archipelago; (2) west of the island Kastellholmen in Stockholm Harbour; (3) southern Blekinge Archipelago; (4) south east
of Tvärminne Zoological Station in Southern Finland Archipelago (Fig. 1). Only a small subset of the acquired multibeam bathymetry is shown from the surveyed area in southern Blekinge Archipelago, and without geographic coordinates, due to military restrictions of revealing detailed depth information. Geophysical mapping results from all other surveyed areas in this study are granted permission to be shown in their full extent and resolution.

### 2.2 Geophysical mapping, processing and analyses

The high-resolution multibeam bathymetric data shown here were acquired using Stockholm University research vessels RV *Electra* (24.3×7.2 m) and RV *Skidbladner* (6.4×2.4 m). A Kongsberg EM2040 0.4°×0.7°, 200-400 kHz, multibeam echo-sounder is hull mounted in RV *Electra* while RV *Skidbladner* has a pole-mounted Kongsberg EM2040 0.7°×0.7°, 200-400 kHz multibeam. Both RV *Electra* and RV *Skidbladner* receive position, heading and attitude data from Kongsberg-Seatex





Seapath 330+ navigation units with attached MRU5+ motion and reference sensors. The Seapath 330+ uses both GPS and GLONASS satellites and is capable of applying real-Time Kinematic (RTK) corrections, which for all surveys in Swedish waters were received from SWEPOS (https://swepos.lantmateriet.se/) over the internet. In Finland, RTK corrections were provided over the internet by the Finnish equivalence, FinnRef (https://www.maanmittauslaitos.fi/en/node/1881). The

Seapath-systems on both vessels generally indicate positions with horizontal accuracies better than ±5 cm and slightly coarser vertical accuracies. A Valeport MiniSVP sound velocity probe was used at discrete stations to acquire sound speed profiles in all surveyed areas for processing of the multibeam bathymetry. In addition, both RV *Electra* and RV *Skidbladner* have AML sound velocity probes mounted near the transducers for continuous reading of surface sound speed.

Sub-bottom profiles were acquired with RV *Electra* using the hull-mounted Kongsberg Topas PS40, 24 channel, parametric sub-bottom profiler operating with primary and secondary frequencies of 35-45 kHz and 1-10 kHz respectively. The sub-bottom profile shown from the eastern part of surveyed area east of the island Askö was acquired using the system in chirp mode with a 4 ms long 4-10 kHz pulse. This system receives positions from the Seapath 330+. The second sub-bottom profile shown from the Askö area, located along the island, was collected in 2009 with a towed Edgetech SB-216S chirp sonar operated

with a 1 ms long 3-9 kHz pulse. Positions were received from a Hemisphere VS100 GPS augmented with SBAS, yielding horizontal accuracies better than ±2 m. However, since the SB-216S was towed at about 3 m water depth, georeferencing of the profile is considerably less accurate. The sub-bottom profile shown from Stockholm Harbour was acquired using a Kongsberg EA640 and a towed surface device, where an Air15-17 transducer with a centre frequency of 15 kHz was mounted 20 cm below the water surface. A 13-17 kHz, 1.024 ms long, chirp pulse was used. Positions were received from a Hemisphere

VS100 GPS with the antenna mounted on the towing device directly above the transducer to avoid offsets. There were no sub-bottom profiles collected from the area in Blekinge, southern Sweden, surveyed with RV *Skidbladner* in 2012. The terraces shown from the area east of Tvärminne in southern Finland were not captured properly with sub-bottom profiles during the two RV *Electra* field campaigns 2017 and 2018 because they are located close to shoals, which prevented a profile from being captured perpendicular to the terraces' extensions.

In 2013, we towed a Klein 3000 100/500 kHz side-scan sonar over a stretch of seafloor with identified terraces east of the island Askö. Positions were received from a Hemisphere VS100 GPS augmented with SBAS, i.e. a navigation setup similar to the SB-216S sub-bottom profiles described above.

Post processing of the multibeam bathymetry was done using the QIMERA software by QPS, version 1.7.2 (https://www.qps.nl/). Bathymetric grids were produced in QIMERA with resolutions ranging from 1×1 m to 0.25×0.25 m. These grids were subsequently imported to the Open Source Geographic Information System QGIS, version 3.6.0-Noosa





(QGIS Development Team, 2018), for further analyses and map making. The sub-bottom profiles were analyzed using the Open Source software OpendTect, version 6.2.0 by dGB Earth Sciences (https://www.dgbes.com/). However, the sub-bottom profile images shown in this article were produced using software tools provided by the Geological Survey of Canada, courtesy Bob Courtney. Flares were identified in the multibeam midwater data, logged in the area east of Askö, using the FMMidwater

suite of Fledermaus software. The "flare hunter" plug-in was used to batch process the data from individual days and identified flares were manually spot checked and subsequently exported as a series of geo-referenced points for production of maps in QGIS.

## 2.2 Bottom inspection and photography

In the Askö and Tvärminne areas, terraces were photographed and filmed using two different Remotely Operated Vehicles

(ROVs): (1) Saab Seaeye Falcon; (2) BlueROV2 by Blue Robotics. The photos shown from Stockholm Harbour and Blekinge were captured by divers. A 3D-model over an approximately 6 m long stretch of the terrace in Stockholm Harbour was assembled using the Agisoft Metashape software (https://www.agisoft.com/).

## 2.3 Sediment coring, measurements and analyses

Numerous sediment cores have been retrieved east of Askö from the areas with terraces during a field component of a

Stockholm University course in marine geophysical mapping. This course has used Askö yearly for field work from 2009 to 2018. In Tvärminne, the mapped terraces were cored in 2017 and 2018. We present analyses of core Asko2018HT-2GC, retrieved at 58°50.76'N 17°41.94'E in 16 m water depth east of Askö, because it illustrates the characteristic sediment of the terraces we cored. The core was taken with Stockholm University piston/gravity coring system adapted for RV *Electra*. This corer is capable of handling a core head with a maximum weight of 563 kg. The 6 m long core barrel is loaded with PVC liners

having outer/inner diameter of 110/98.5 mm. Core Asko2018HT-2GC was retrieved with the corer rigged in gravity mode.

## 2.4 Sediment physical properties

Core Asko2018HT-2GC was subjected to high-resolution (1 cm) logging of sediment physical properties, including bulk density, magnetic susceptibility and p-wave velocity, using a Geotek Multi-Sensor Core Logger (MSCL). The 4.3 m long core

was cut into 1.5 m long sections that were logged before being split, described and digitally imaged. Sediment grain size (0.2 µm–2 mm) was measured on 29 samples using a Malvern Mastersizer 3000 laser diffraction particle size analyzer. Wet samples were immersed in a dispersing agent (<10% sodium hexametaphosphate solution) and placed in an ultrasonic bath to facilitate particle disaggregation before analyses. Particle classes were defined using the European Standard EN ISO 14688-1:2018 where clay is the 0–2 µm fraction, silt 2–63 µm and sand >63 µm.



## 2.5 Geochemical analyses

A carbonate concretion found in core Asko2018HT-2GC was subjected to geochemical analyses at the Stabile Isotope Lab of the Department of Geological Sciences, Stockholm University, for determining the water source(s) during its formation. Two pieces of the concretion were analysed: one from the centre and one from the outer edge. The samples were milled to a powder

and analyzed for total carbon using a Termo Delta V mass spectrometer and for phosphoric acid reactive carbon with a Gasbench II-MAT253 mass spectrometer. The elemental analysis provides $\delta^{13}C$ of the total carbon while the Gasbench II provides $\delta^{13}C$ and $\delta^{18}O$ of the carbonate. Paired t-tests were applied to address if the two methods provided the same $\delta^{13}C$ results, while t-tests were used to test if there were significant differences between the results from analysing samples from the inner respective outer parts of the concretion. A confidence interval of 95 % were applied in all t-tests. The $\delta^{13}C$ and $\delta^{18}O$

are reported with respect to the Vienna Pee Dee Belemnite (VPDB) standard (Coplen, 1996).

## 3 Results

### 3.1 Geophysical mapping

Terraces in the seafloor were mapped by Jakobsson et al. (2016) using a multibeam echo-sounder in the vicinity of the island Skåren, located east of Askö (Fig. 2). These terraces have steps of about 1 m high, extend from a few meters to >100 m in length, and are most abundant in water depths shallower than 15 m. In addition, a few semi-circular depressions were mapped that resemble pockmarks with about 1 m bathymetric expression, although with an opening towards the down-sloping seafloor

(Jakobsson et al., 2016). Their semi-major axes are between 10 and 30 m. Here we have expanded the mapped area east of Askö considerably east- and northward and derived descriptive statistics of the terraces' depth distribution (Fig. 2). Most terraces are located in 12 m water depth while the mean and median depths are 16 m and 15 m respectively. The shallowest depth of a mapped terrace is 4 m while the deepest is 28 m. However it should be noted that our mapping did not extend much shallower than about 4 m water depth. Many of the terraces extend laterally in rather systematic winding sinusoidal patterns,

although there are some that take the form of a single hyperbola (Figs. 2c and d). The height of the terrace steps varies, but is generally <1 m (Fig. 2e).

The multibeam bathymetry shows that the seafloor morphology of the terraces in the area east of Askö, Tvärminne and southern Blekinge Archipelago closely resemble one another (Figs. 3 and 4). There are far fewer mapped terraces in the latter two areas



preventing a meaningful statistical comparison of their depth distribution. However, the terraces we mapped outside of the area of Askö occur in the deeper depth range: (1) Tvärminne 15-20 m; (2) Blekinge Southern Archipelago 23-25 m. The terrace in Blekinge is the longest we mapped, it is possible to trace continuously for >550 m (Fig. 4a). We do not have multibeam bathymetry of the terrace found in Stockholm Harbour preventing a comparison of its morphology and spatial extent to the

terraces in the other three areas.

Multibeam water column information was logged and analyzed for two-thirds of the surveyed area east of Askö. Seeps from the seafloor were found to be a common feature (Fig. 2a). There is an abundance of seeps in the northern part where no terraces are identified. East and southeast of the island Skåren, seeps begin to occur at about 20 m water depth, i.e. with a few exceptions

this is from where the deepest terraces occur (Fig. 2b). The mean and median depths of the seeps are 21 and 20 m respectively, while the shallowest is located in a water depth of 3 m and the deepest in 40 m. The multibeam backscatter shows that the terraces in the area east of Askö systematically appear in a relatively harder seafloor characterized by high backscatter while the seeps generally occur in a softer seabed represented by lower backscatter (Fig. S1 in the Supplement). The side-scan data acquired in 2013 show high-resolution imagery of the terraces, with no apparent difference in signal intensity across them, but

with clear shadows present due to the bathymetric expressions (Fig. S2 in the Supplement).

There is a semi-regular grid of sub-bottom profiles covering the entire area east of Askö (Fig. S3 in the Supplement). These profiles were acquired during the field component of a Stockholm University course in marine geophysical mapping held yearly in this area since 2009. From this database two sub-bottom profiles, extending across terraces identified in the multibeam

bathymetry are shown in Figures 5a and b. The terrace west of the island Kastellholmen in Stockholm Harbour was mapped by a sub-bottom profile perpendicular to its extent (Fig. 5c). Common for all terraces imaged by sub-bottom profiles is that the acoustic stratigraphy indicates well-stratified sediments that outcrop at the seafloor where the terrace is formed. This is particularly clear in the profile from Stockholm Harbour (Fig. 5c). In the area east of Askö, where the sub-bottom profile coverage is most comprehensive, an acoustically semi-transparent surface unit with few internal noticeable reflections is

commonly found in sections with water depths deeper than the general occurrence of terraces (Fig. 5b).

## 3.2 Bottom photography

The bottom photos confirm that distinct terraces are formed in the seafloor in all mapped areas (Figs. 5d-f, 6a-f.). It is also possible to identify that the terraces developed in stratified sediments outcropping at the seafloor, which was particularly

evident in the acoustic stratigraphy of the sub-bottom profile from Stockholm Harbour (compare Fig 5c with Figs 5e-f). Holes with diameters between about 1 to 2 centimetres are abundant in the near vertical terrace walls in Stockholm Harbour and



Tvärminne (Figs. 5d,e and 6c,d). Some of these holes appear to be cavities from stones that were embedded in the sediments and eventually fell out during the erosional process forming the terrace. However, we cannot confirm if this is always the case because the holes sometimes appear to extend rather deep into the terrace walls. Hence they may be zones of piping and erosion that developed in response to focused groundwater flow.

## 3.2 Sediment stratigraphy and physical properties

The 4.3 m long core Asko2018HT-2GC, retrieved from a terrace in a water depth of 16 m in the area east of Askö, consists of rhythmically alternating 0.5-2.5 cm thick silty-clay layers (Fig. 7). The upper 3.5 m is composed of inclined (~10-16°) rhythmites with a gradual transition into the lower 0.8 m thick interval with near horizontal rhythmites. Micro-faults offsetting

some rhythmites by ~0.5 cm appear in the core from about 1.5 m core depth and are present to the bottom of the core. The silt content within the 0.5-2.5 cm thick rhythmites varies as well as the colour, which changes from greyish brown (Munsell: 5/2 10YR) to reddish brown (Munsell: 5/3 5YR). Over the interval where grain size measurements were obtained (2.80-4.20 mbsf) the sediments are composed almost exclusively of fine-grained material (>95% is <20 µm) (Fig. 8). At about 3.5 m, p-wave velocity, bulk density and magnetic susceptibility all transition towards slightly higher values, with peak values occurring in

zones slightly enriched in silt (Fig. 7). Sediment porosity, calculated from the bulk density logs by assuming a grain density of 2.71 g/cm³, also decreases from ~60-70% to ~55-60% below 3.5 m. No disconformity is recognised across this transition. The porosity reduction does not appear to be driven by normal downhole compaction, but may be related to the slight coarsening of the sediments.

Two carbonate concretions were found at core depths of 1.77 m (diameter 1 cm) and 2.64 m (diameter 5.3 cm) (Fig. 7). The lowermost one has a disc-shaped appearance, displaying a series of concentric rings growing outwards from a central spherical concretion. It is a classic *Marleka*, a.k.a. 'fairy-stone' or 'imatra stone', which form in Quaternary clay-rich sediments as calcium carbonate precipitates around a small pebble or organic matter (Neuendorf et al., 2011).

## 3.4 Geochemistry

The results from the geochemical analyses on the concretion are shown in Table 1 ($\delta^{13}$C and $\delta^{18}$O are reported in ‰ versus VPDB). The elemental analyses of the outer edge of the concretion give an average $\delta^{13}$C of -20.23 ‰ (SD=0.11), while the Gasbench-II yields and average of -19.87 ‰ (SD=0.02). A paired t-test (p=0.04) indicates a significant difference between the two methods for the outer edge samples considering a 95 % confidence interval (alpha=0.05). From the central sample, an average of -19.35 ‰ (SD 0=22) and -19.32 ‰ (SD=0.03) were provided by the elemental analyses and Gasbench-II methods

respectively. A paired t-test (p=0.88) suggests that there is no significant difference between the measured centre samples





using the two different approaches. If we compare the average of all measurements made on the samples from the edge of the concretion (M=-20.05 SD=0.21) with the average of all measurements made on the samples from the centre (M=-19.33 SD=0.14), a t-test suggests a significant difference (p=0.00) in $\delta^{13}$C, albeit small. The average $\delta^{18}$O from the central sample is -10.54 ‰ (SD=0.19) and the edge -8.02 ‰ (SD=0.14), and a t-test suggesting a significant difference (p=0.00) between the

central and outer edge of the concretion with respect to $\delta^{18}$O. The total carbon has an average of 6.98 % (SD=0.36) at the edge and 10.46 % (SD=0.32) in the centre. A t-test (p=0.00) gives that these means are significantly different from each other.

## 4 Discussion

The first discoveries of terraces formed in the seafloor along the Swedish coast were made in the 1990s with a conventional
30 kHz echosounder and a 100/500 kHz side-scan sonar (Fig. 1; Söderberg and Flodén, 1995). Our detailed surveys east of the island Askö showed that for a terrace to be identified with a single beam echo sounder, the profiling direction cannot deviate much from being perpendicular to the orientation of the terrace. The main reason for this is that the bathymetric expressions of the terraces rarely exceed 1 meter. They are easier to map with side-scan sonar, although as no bathymetry is provided by a conventional side-scan sonar, they may also be misinterpreted as other surface patterns in the seafloor sediments (Fig. S2 in
the Supplement). It is only with the latest generation of shallow water high-resolution multibeam echo sounders, that the terraces are irrefutably recognized as prominent bathymetric features in the seafloor (Figs. 2-4). This may explain why it took around two decades to realize how common the type of terraces mapped by Söderberg and Flodén (1995) are along the Swedish coast, as well as elsewhere along the coasts of the Baltic Sea. Recognizing that the features are widespread, the hypothesized formation mechanism involving SGD makes them important from an environmental point of view. The terraces could comprise
focal points where groundwater enters the Baltic Sea to influence its brackish waters by providing, not only freshwater, but potentially pollutants and nutrients. If this is the case, quantification and chemical analyses of the SGD from the terraces will provide new information for assessments of the nutrient budget of the Baltic Sea.

Our study supports the hypothesis put forward by Söderberg and Flodén (1995) on the terrace formation mechanism, i.e.
groundwater flows through siltier layers in glacial clay to eventually escape at the seafloor where erosion from the flowing water undermines the layers above so they collapse to form a sharp terrace in the seafloor (Fig. 9). The photos in Figure 6 include examples where we believe that this process can be readily envisioned. It is possible to see how cavities are formed in the varved clay at the bottom of some of the terraces as well as blocks of the overlying clay that have collapsed from being undermined.




We have also found from our geophysical mapping and coring results that the terraces are systematically formed in glacial clay throughout the studied areas. Glacial clay in the Baltic Sea sediment stratigraphy is commonly found draped on top of till or glaciofluvial material, or in some cases, rests directly on bedrock (Andrén et al., 2011; Andrén et al., 2015) (Fig. 9). This type of clay was for the most part deposited during the last deglaciation in front of the retreating Scandinavian Ice Sheet.

Glacial clays left from previous older glaciations are extremely rare and found only at a few locations (e.g. Björck et al., 1990). Swedish geologist Gerard De Geer discovered that glacial clay is comprised of rhythmites, where the layers composed of varying proportions of clay and silt are annual depositions of erosional material from the retreating ice sheet (De Geer, 1912). He introduced the term "varve" for one annual layer of glacial clay and noted that its thickness and silt content varied depending on the proximity to the retreating ice margin, i.e. thicker and siltier varves were deposited close to the ice margin. He further

proposed that there would be a higher degree of silt content in the part of the varve representing the meltwater rich summer period. Thus, grain size variations in the glacial clays are found on a number of scales, from mm to cm scale variations across rythmites, to longer (decimetre) scale variations related to climatically driven variations in subglacial discharge, to even longer (> meter) scale variations related to the proximity of the ice margin. From this knowledge follows that there is a higher chance of finding more silt-rich layers in the older sections of glacial clay deposited close to the ice margin. This would be in the

lower sections of the glacial clay units. In the case of Asko2018HT-2GC, this larger scale variation appears to be captured in the coarsening of the grain size below 3.5 m depth, which has length-scales that exceed the duration of individual rhythmites (Fig. 7). Although fine-grained glacial clays are not commonly considered to be highly transmissible sediments, variations in the silt content of clay-rich sediments has a dominant effect on their permeability (Schneider et al., 2011). Experimentally, it has been shown that at a given porosity for silt/clay mixtures, increasing the clay content (<2 μm) from 36 % to 57 % decreased

the permeability by an order of magnitude (Schneider et al., 2011).

It should be noted that the glacial clay sequences are time transgressive throughout the Baltic basin, with older clay in the south and younger in the north because the Baltic basin was deglaciated from south to north (Hughes et al., 2016). The first

small freshwater body in which glacial clay could be deposited during the last deglaciation, formed in front of the ice margin around eastern Denmark and the northern coasts of Germany and Poland at about 16-15 ka BP (Houmark-Nielsen and Henrik Kjær, 2003). This water body grew as the ice sheet retreated northward to become the Baltic Ice Lake, which appears to have lasted until the end of the Younger Dryas Cold period at about 11.7 ka BP, when it catastrophically drained westward north of Mount Billingen in south central Sweden (Björck and Digerfeldt, 1986; Andrén et al., 2002; Swärd et al., 2015). Mapping of

preserved paleo-shorelines in the 1920s had showed that the drainage occurred when the ice sheet margin reached north of the damming high terrain in the west (Lunqvist, 1921). A brackish water phase called the Yoldia Sea followed the Baltic Ice Lake (Björck, 1995), however it would take several hundred years for the Baltic to become brackish after the drainage and deposition





of varved glacial clay continued close to the retreating ice margin (Andrén et al., 2011). The Baltic Sea basin was completely ice free at about 10 ka BP (Hughes et al., 2016).

The geochemical analyses of the concretion from core Asko2018HT-2GC, do not irrefutable determine whether or not

groundwater has flown through the glacial clays, but provide valuable insights into the formation environment. The $\delta^{13}$C carbonate isotope values of the concretion between about -19 and -20 ‰ are low compared to Baltic Sea $\delta^{13}$C DIC (Dissolve Inorganic Carbon), which is usually between 0 and 1 ‰ (Filipsson et al., 2017). Carbonates formed from Baltic Sea water should thus have similar values. Therefore the most plausible explanation for the observed isotope values is that the carbon source is respired organic matter in the sediments. Organic matter in the Baltic Sea is usually in the range -23 to -28 ‰

depending on if it is coming from terrestrial sources or from primary production in the Baltic Sea (Alling et al., 2008; Deutsch et al., 2012). The small difference in isotope values between the concretion and organic matter in the Baltic Sea could be due to mixing with DIC from Baltic Sea water during formation.

The $\delta^{18}$O values of samples taken from the edge and the centre of the concretion are significantly different from one another

by about 2.5 ‰. Here we investigate possible causes for the isotopic differences between these samples using the Shackleton (1974) equation for relating temperature with $\delta^{18}$O$_c$ and $\delta^{18}$O$_w$

$$t = 16.9 - 4.0(\delta^{18}O_c - \delta^{18}O_w) \qquad (1)$$

where $t$ is temperature, O$_c$ is from carbon dioxide extracted from carbonate and O$_w$ is from carbon dioxide equilibrated with

water. From Equation 1 follows that the $\delta^{18}$O$_w$ could be different in the samples from the edge compared to the samples from the middle if the formation temperatures were different. However, the difference between the middle and the centre is too large to be explained alone by a change in temperature during the formation. It more likely indicates a change in composition of the water source, perhaps in a combination with a temperature change.

The precipitation in the Stockholm area has a yearly mean $\delta^{18}$O of about -10 ‰ (the Global Network of Isotopes in Precipitation, GNIP database, IAEA/WMO). The Baltic Sea has currently a $\delta^{18}$O value between about -8 and -9‰ at the salinity where the core is taken, east of the Island Askö (Deutsch et al., 2012). However, the Baltic Sea stages during which glacial clay were deposited had most likely much lower values considering the influence from the meltwater of the Scandinavian Ice Sheet. For example, isotopic $\delta^{18}$O values of the Greenland Ice Sheet are in the order of -35 to -40 ‰ (Andersen et al., 2004).

This is in line with isotopic measurements made on pore water from a sediment core retrieved in Lake Vättern, which show a prominent down-core progression to $\delta^{18}$O values of less than -35 to -40 ‰ in the sections of the core representing the Baltic Ice Lake and older (Fig. S4 in the Supplement). This 74 m long sediment record captured the Baltic Sea stages from the Baltic



Ice Lake to the time Lake Vättern was isolated 9530 ±50 cal years BP (Swärd et al., 2018). The generally low $\delta^{18}O$ values from the concretion (-7.93 to -10.70 ‰) are much less negative than modern Baltic Sea water, and even further removed from the inferred $\delta^{18}O$ composition of the Baltic Ice Lake (< -35 to -40 ‰). This suggests that groundwater took part in the formation of the concretion, because if it would have been water from a Baltic Sea stage heavily influenced by meltwater, lower $\delta^{18}O$

values should be expected.

Isostasy plays an important role in estimating the potential hydrogeological connection in the Baltic region between land and sea through glacial clay. The Swedish terrain was isostatically depressed by the several kilometres thick Scandinavian Ice Sheet (Lambeck et al., 2010) and the highest coastline is therefore found in several areas far inland of the present coast

(Björck, 1995). Isostatic rebound eventually caused glacial clay sequences to be lifted above the level of the Baltic Sea, thereby creating a hydraulic head between the landward end of permeable glacial clay layers and their seaward continuation (Fig. 9).  To estimate SGD into the Baltic Sea through these permeable layers we can apply Darcy's law according to

$$Q = -KA\frac{dh}{dL} \qquad (2)$$

where $Q$ is the water flow (m³/s) in a saturated porous material, $K$ is the hydraulic conductivity (m/s), $A$ cross section area

(m²), and $dh$ and $dL$ the difference in height (pressure drop) with respect to distance over which the water flows. Future estimates of the SGD rates into the Baltic Sea from permeable layers in glacial clays, will require knowledge on where terraces occur on the seafloor, the hydraulic head, i.e. $dh$ and $dL$, and the hydraulic conductivity $K$ of the permeable siltier layers, which is governed by their composition. Such estimates will require high resolution mapping efforts to identify potential locations of SGD, cross-sectional areas, and emission depth.

The SGD through the terraces will most likely be intermittent considering the relatively large variation of the seasonal ground water table around the Baltic Sea, as shown by the geological surveys' monitoring on land. Looking at data from monitoring stations provided by the Swedish Geological Survey in the vicinity of the Stockholm Archipelago, we note that the seasonal variation in the groundwater table of some locations exceeds 3 m and the highest groundwater table is generally found from

the late fall to late spring (Fig. S5 in the Supplement). Groundwater discharges to the Baltic Sea through the seafloor are known to occur through geological formations other than the terrace formations discussed here, in particular where glaciofluvial aquifers on land connects with the seafloor (Peltonen, 2002). However, the prevalence of the SGD related terraces in the mapped regions suggests that SGD to the Baltic Sea is likely underestimated and remains an unconstrained source of pollutants and nutrients, as previously argued by (Destouni et al., 2008). The ecosystem role of the terraces is therefore unknown. It is

also clear that there is a general lack of data and understanding of SGD emission mechanisms for proper assessments of emission rates and/or volumes (Taniguchi et al., 2002).



**Data availability**

The multibeam bathymetry and sub-bottom profiles presented in this work have been granted public release by the Swedish Maritime Administration, apart from the data shown from Blekinge without coordinates and precise location information. The released data are available for download from the Bolin Centre Database: https://bolin.su.se/data/ .

5  **Author contribution**

M. Jakobsson prepared the manuscript with input from all co-authors. M. Jakobsson analysed and processed the multibeam bathymetry and sub-bottom profiles, M. O'Regan measured and analyzed the sediment cores, C.-M. Mörth did the geochemical analyses and E. Weidner identified seeps in the multibeam water column data. J. Hansson and A. Sjöström captured photos of terraces while diving.

**Competing interests**

The authors declare that they have no conflict of interest.

**Acknowledgements**

15  We thank the crew and Captain of RV *Electra* and the Baltic Sea Centre for their support. M. Jakobsson worked on this paper during sabbatical leave supported by Stockholm University and thank NIWA (National Institute of Water and Atmospheric Research) in Wellington, New Zealand, for providing a work space during the sabbatical. Dr Geoffroy Lamarche at NIWA is specifically thanked for fruitful discussions. Further funding was provided by The Academy of Finland (project ID 294853), and the University of Helsinki and Stockholm University strategic fund for collaborative research (the Baltic Bridge initiative).

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



**Table 1:** Geochemical analyses of the concretion found in the glacial clay unit of core Asko2018HT-2GC. See Figure 2a for core location and Figure 7 for an image of the concretion. The upper section are elemental analyses made using the Termo Delta V mass spectrometer and the lower analyses made on phosphoric reactive carbon with a Gasbench II-MAT253 mass spectrometer (see methods for further details).

**Figure 1:** Overview map showing the locations of the studied areas in this work with white dots. Corresponding figures for each area are shown with white text. The locations of previous studies showing SGD discussed here are shown with black dots. I/II=Söderberg and Flodén (1995); III=Schlüter et al. (2004); IV=Virtasalo et al. (2019). The bathymetry is from EMODnet 2018 (EMODnet Bathymetry Consortium, 2018)

**Figure 2.** Multibeam bathymetry of the studied area east of the island Askö, southern Stockholm Archipelago. (a) Overview map with brown lines representing digitized terraces in the seafloor and blue dots showing locations where seeps were identified in the multibeam water column data. Black heavier lines show the locations of the sub-bottom profiles in Figures 5a (SBP1) and 5b (SBP2). The sub-bottom profiles in Figure 5 are portrayed between the two dots, with the profile running from left (black dot) to right (white dot). The location of core Asko2018HT-2GC is marked by a yellow star (abbreviation 02GC used on the map). (b) Depth statistic of the occurrence of terraces and seeps. (c) Detailed view of the terraces in the seafloor around the island Skåren. (d) The location of the bathymetric profile across two terraces from X to X´ shown in (e).

**Figure 3.** Multibeam bathymetry of the studied area east of Tvärminne Zoological Station in Southern Finland Archipelago. (a) Overview of the seafloor bathymetry in the vicinity of the only set of terraces mapped in this area, shown in (b). (c) Bathymetric profile from Y to Y' across two terraces. The profile location is shown in (c).

**Figure 4.** Multibeam bathymetry of a terrace mapped in southern Blekinge Archipelago. (a) Only a smaller part of the full survey is shown and without coordinates, because detailed bathymetric information from this area is under military restrictions. (b) Detail of the >500 m long terrace with the location of the bathymetric profile from Z-Z' shown in (c).

**Figure 5.** Sub-bottom profiles and bottom photographs portraying seafloor terraces. (a,b) Sub-bottom profiles SBP1 (a) and SBP2 (b) from the area east of Askö. The locations of terraces clearly identified in the multibeam bathymetry are marked T1-T9 (see Figure 2a). The site of core Asko2018HT-2GC (02GC on profile SBP1) is shown with a black arrow. Note that it is located on top of terrace T1. (c) Sub-bottom profile across the terrace mapped west of the island Kastellholmen in Stockholm Harbour. Bottom photographs and 3D-photo mosaic of this terrace are shown in (d)-(f).

**Figure 6.** Bottom photographs of seafloor terraces in the studied areas. (a,b) East of the island Askö, (c,d) east of Tvärminne Zoological Station in Southern Finland Archipelago, (e,f) near Boön, Blekinge Archipelago.

**Figure 7.** Lithology and sedimentology of sediment core Asko2018HT-2GC. Image includes a lithologic log (a), sediment physical properties measured on the MSCL (b), and results from grain size analyses showing variations in clay and silt content across the zone where the overall porosity reduction is seen (~ 3.5 m). Characteristic images from four different depths are shown (d), including the 2 cm diameter concretion recovered at 2.64 m and the smaller concretion at 1.77 m.

**Figure 8.** Cumulative grain size distributions for the laminated glacial clays in Asko2018HT-2GC. Sediments are exclusively composed of fine fraction material, with 50-80% of the sediments being <4 μm.

**Figure 9.** Schematic illustration showing how siltier layers in glacial clay could act as conduit for ground water, eventually escaping at the seafloor as SGD, leading to the formation of a bathymetric terrace. The critical parameters $dh$ and $dL$ in Darcy's law (Eq. 2) are illustrated in the sketch as they dictate the hydraulic head that would drive the flow.





Table 1.

Elemental Analyses (Termo Delta V)

| Sample | δ¹³Ctot vs VPDB (‰) | average δ¹³Ctot | SD | % Ctot | average % Ctot | SD |
|---|---|---|---|---|---|---|
| Edge 1 | -20.30 | **-20.23** | **0.11** | 7.04 | **6.97** | **0.36** |
| Edge 2 | -20.11 | | | 7.30 | | |
| Edge 3 | -20.29 | | | 6.59 | | |
| Centre 1 | -19.54 | **-19.35** | **0.22** | 10.19 | **10.46** | **0.32** |
| Centre 2 | -19.39 | | | 10.37 | | |
| Centre 3 | -19.10 | | | 10.81 | | |

Acid reaction analyses of carbonate (Gasbench II-Mat253)

| Sample | δ¹³C vs VPDB (‰) | δ¹³C average | SD | δ¹⁸O vs VPDB (‰) | δ¹⁸O average | SD |
|---|---|---|---|---|---|---|
| Edge 1 | -19.86 | **-19.87** | 0.02 | -7.93 | **-8.02** | **0.14** |
| Edge 2 | -19.89 | | | -8.18 | | |
| Edge 3 | -19.84 | | | -7.94 | | |
| Centre 1 | -19.35 | **-19.32** | 0.03 | -10.32 | **-10.54** | **0.19** |
| Centre 2 | -19.28 | | | -10.60 | | |
| Centre 3 | -19.34 | | | -10.70 | | |





Figure 1.







Figure 2.



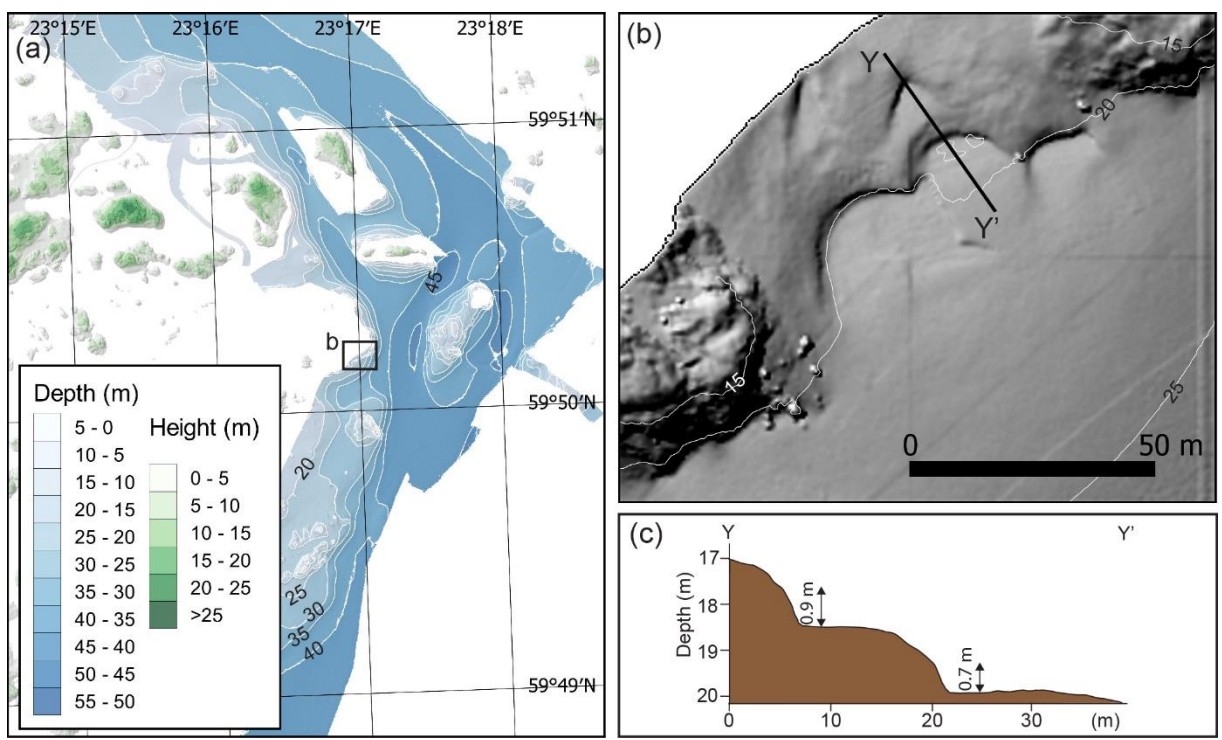

Figure 3.



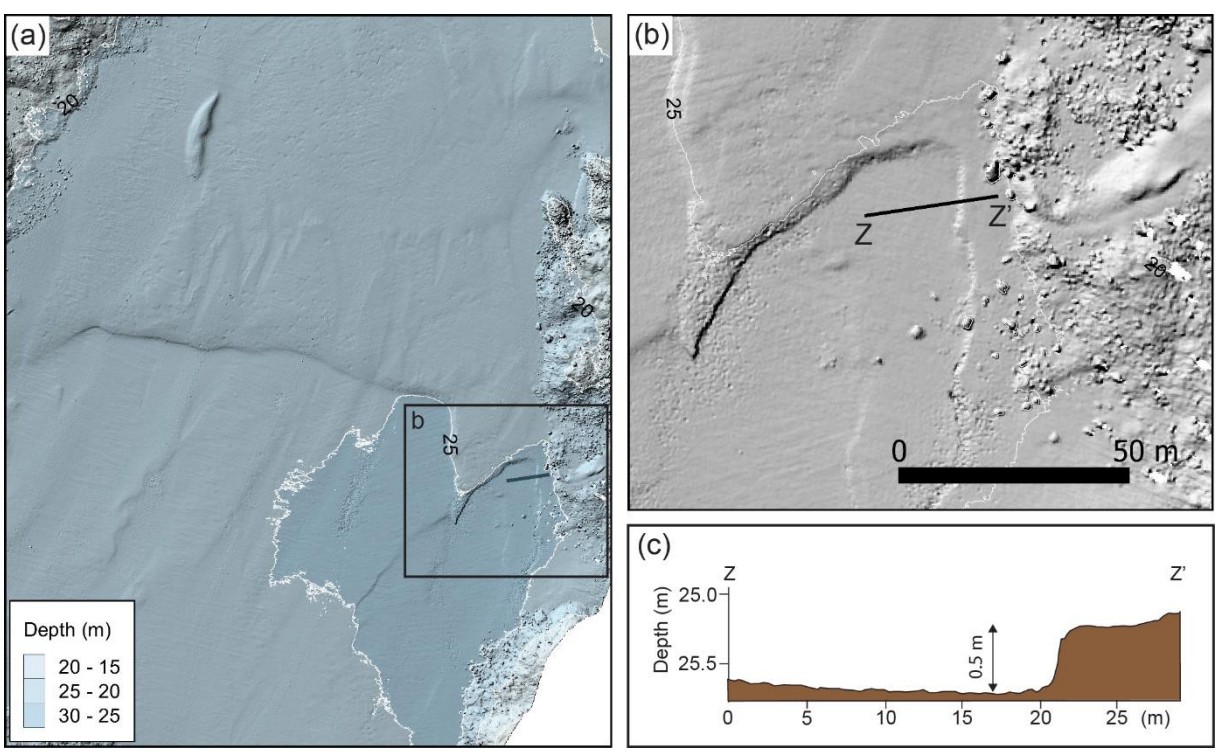

Figure 4.





Figure 5.





5    Figure 6.

Earth **Surface**
**Dynamics**
Discussions





Figure 7.





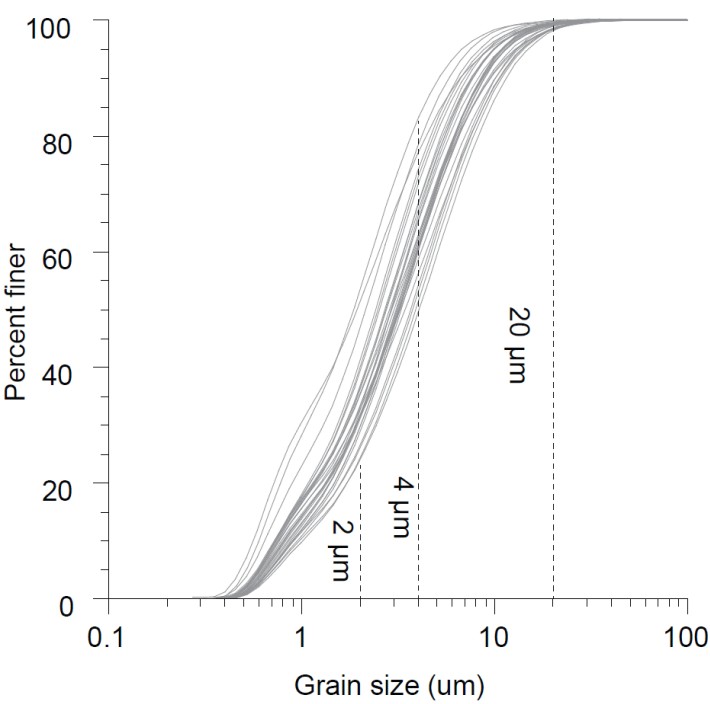

Figure 8.



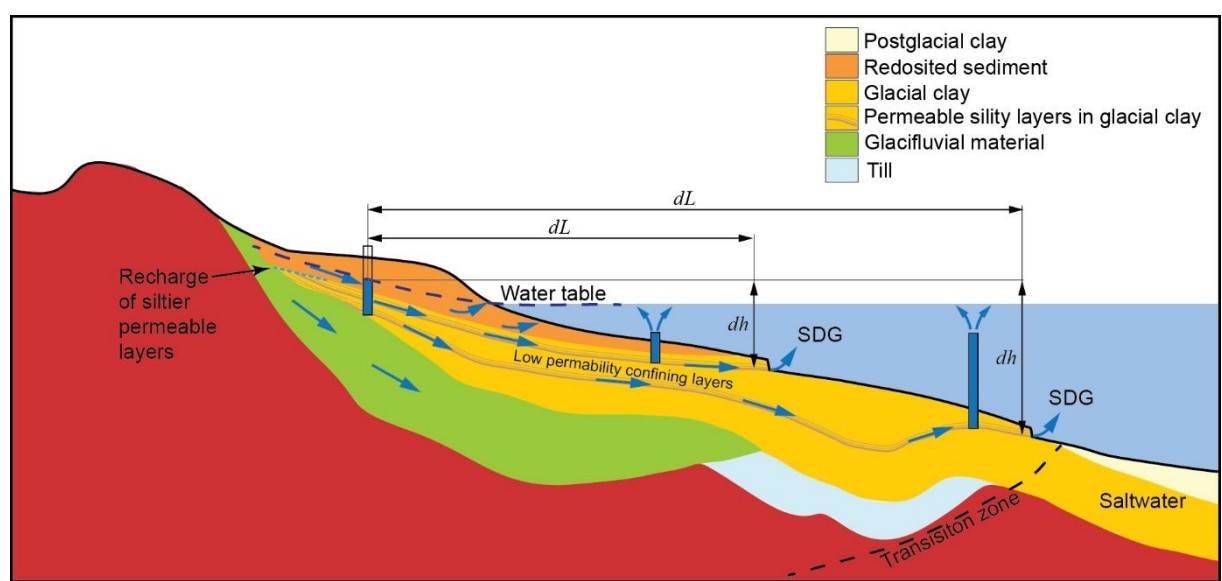

Figure 9.