# Peer review of "Potential links between Baltic Sea submarine terraces and groundwater seeping"

_Earth Surface Dynamics, 2019_

## Referee Comment (RC1) · Anonymous Referee #1 · 4 Sep 2019

Firstly, I find the word "sapping" a bit problematic since that word has many meanings. Have you considered "seepage"? I suppose you then will miss the under-mining part of the process, but…..? In general I find this to be a good manuscript on an almost neglected topic. It is also free from erroneous or speculative statements, but presents very reasonable interpretations and gives a good overview on the importance of combining the detailed submarine bathymetry, lithostratigraphy and hydrogeology to understand the under-lying processes. However, I think it would add an interesting aspect to the paper with a discussion on how long this "sapping"process has been going on and how it may be related to the Baltic Sea development and local isostasy. Are there differences between Finland, Stockholm and Blekinge? Are the right submarine conditions only met in areas with a continuous regression since deglaciation? One

could for example speculate that the Yoldia Sea low stand below today's sea level in SE Sweden (and not e.g. in Stockholm), with the regressions and transgressions of the Yoldia-Ancylus-Littorina phases, reworked the coastal and shallow marine varved clay sequences to such a degree that you would usually not get the right conditions for "sapping" above -15 to – 20 m in Blekinge (which is the appr. the low stand level there), in contrast to the Stockholm area where todays submarine varved clay units have been below 0 m throughout the Holocene. So I only have some very minor points to comment and warmly recommend publication. p. 6 l. 2: I suppose it should be Stable, not Stabile? p. 6. l. 22: How can most of the terraces be in 12 m water depth when the mean depth is 16 m? Are there so few in shallow depths/many in deeper depths? p. 10 l. 24: Why use the reference Hughes et al. (2016) when there are so many local/regional studies of the deglaciation of the Baltic Sea? p. 11 l. 32: it says "Ice Lake and older" but what is older than the Baltic Ice Lake in the Lake Vättern core? The thick varved clay sequence is most likely of Baltic Ice Lake origin.

---

## Referee Comment (RC2) · Joonas Virtasalo (Referee) · 29 Sep 2019

General comments

This manuscript presents a study of seafloor morphological features in Stockholm Archipelago and Blekinge Archipelago in Sweden, and an archipelago area in the southern Finnish coast, focusing on submarine terraces that are interpreted to be formed by groundwater sapping. The study is based on high-resolution multibeam echosounder bathymetry, supplemented by seafloor photography and filming by ROVs, sedimentological and geophysical analysis of sediment cores, and C and O isotope analysis of a carbonate concretion found in one of the cores. The dataset is strong and has a good regional coverage, the quality of multibeam data is excellent, and

the supporting information is adequate. Overall, the manuscript is well written and illustrated, and the topic is both relevant to the wider audience and suitable for Earth Surface Dynamics. However, parts of interpretation of results and Discussion need to be developed before the manuscript is acceptable for publication. My main concern is that while the authors convincingly explain groundwater sapping as the mechanism of the formation of submarine terraces, they do not discuss other mechanisms that can produce similar structures. The submarine terraces are observed in glacial varved sediments, where liquefaction of the coarser couplet layers is interpreted to have resulted in the detachment and sliding of sediment blocks, leaving behind the observed terrace structures. While groundwater flow can cause liquefaction of the coarser couplet layers, also seismic shaking and reduced hydrostatic pressure as a consequence of rapid water level fall can result in sediment liquefaction and produce similar terrace structures. Paleoseismic activity is well known in the area, and sliding, slumping and debris flow deposition of glacial varved sediments has been previously associated with seismically triggered liquefaction (e.g. Hutri & Kotilainen, 2007, Marine Geology; Virtasalo et al., 2007, Sedimentology). Also drops in water level are known to have taken place previously in the area (e.g. Sauramo 1923, Studies of the Quaternary Varve Sediments of Southern Finland, Bull. Comm. Geol. Finl. 60). The authors thus need to consider other potential mechanisms before they can associate the observed submarine terraces with groundwater discharge as strongly as it is currently written in the manuscript.

My other concern is that the submarine terraces are observed on seafloor areas with relatively low topographic gradients, whereas significant groundwater flow requires a strong hydraulic gradient. How can the required hydraulic gradient be developed in such flat areas and at some distance to the shoreline? Furthermore, in Fig. 5a and 5b, terraces are interpreted on the slopes and tops of local elevations. What would be the mechanism leading to a sufficiently strong groundwater discharge to these local elevations to produce the terrace structures?

The manuscript is lacking hard evidence of groundwater influence such as radon measurements that could potentially help strengthening the interpretation of the formation mechanism of submarine terraces. Isotope analysis of the carbonate concretion is a good attempt in this direction, but the results unfortunately are inconclusive. In case it is not possible to exclude the other possible mechanisms, the tone of the manuscript should be changed from "submarine terraces are produced by groundwater sapping" to "submarine terraces can be produced by several processes, but we interpret our observed structures to be produced by groundwater sapping".

The manuscript also reports seeps from the seafloor, but their connection to terraces, groundwater and the topic of the manuscript in general is not clearly explained.

Specific comments

Page 2, line 25. Add "varved" between "glacial" and "clay". It would help the reader in case the varved clay structure was shortly explained already in this paragraph.

Page 10, the first paragraph about glacial varved clays is very much dominated by Swedish publications. Please consider adding classical works by e.g. Sauramo in the discussion.

Page 10, line 31. There is no compelling evidence for brackish Yoldia Sea northward and eastward from the south-central Sweden (Schoning 2001, Boreas). Yodia Sea is not necessarily relevant to the topic of the manuscript and it could be excluded from the discussion.

Page 11, line 4, "irrefutably".

Page 11, lines 5-12. Perhaps the concretions formed already during glaciolacustrine or post-glacial lacustrine environments, and comparisons to present brackish-water Baltic Sea are not relevant?

Page 12, lines 1-5. How about the O isotope composition of the post-glacial lacustrine phase? Perhaps it was similar to large lakes in Sweden and Finland today?

That ends my referee comments.

**ESurfD**

---

## Author Comment (AC1) · 11 Nov 2019

We thank Referee #1 for constructive and positive comments. The comments and questions raised have been addressed in a revised version of the paper. First, we agree that "sapping" can be a bit problematic to use, even if it commonly occurs in the literature. We therefore follow the recommendation and change the term to "seepage/seeping", also in the title. We agree that the time aspects of the formations indeed are very interesting, however since there are no direct age data available, we do not want to speculate too much. But we have followed the recommendation and included a couple of sentences that raises this topic following the age discussion on the glacial clay deposition on page 11:

[Figure]

"....The Baltic Sea basin was completely ice free at about 10 ka BP (Hughes et al., 2016). As the ice retreated (Stroeven et al., 2016), conditions may have developed for terrace formation at different places around the Baltic Sea depending on local sea level in relation to glacial clay deposits. The mapped terraces in the different regions may therefore be of different ages, some may be inactive while others are active today."

Detailed comments: 1: "Stabile" changed to "stable" in page 6. 2: Statistics of terraces in the Askö area. It was double checked, and come out to median/mean 15/16 m as written in the paper. Referee #1 is correct, it is simply because there are few in the shallower depths, which drives the statistics towards the deeper end. 3: We see the point and have included an additional reference to Stroeven et al. (2016, Deglaciation of Fennoscandia) for the sentence stating how the ice Scandinavian ice sheet retreated over the Baltic basin. 4: Our mistake, should not be "or older"

---

## Author Comment (AC2) · 11 Nov 2019

We thank Referee #2 Joonas Virtasalo for his constructive comments on the manuscript. Virtasalo states that the paper is well written and based on strong data, but he has two main concerns:

a) We do not discuss other mechanisms that can produce similar structures. b) The features in the form of terraces in the seafloor occur in areas of low hydraulic gradient.

We have addressed both these main concerns in a revised version of the manuscript. Concern a) is handled by including other potential mechanisms in the discussion, as requested, at the same time as making it clear that groundwater seeping cannot irrefutably be concluded to be the sole mechanism responsible for producing the sub-

marine terraces we document.

In the paper we describe the morphological features with the new data, place them in a geological context, and address the previously proposed formation mechanism in order to further discuss the formation hypothesis. We do in fact agree with Virtasalo that there is no direct conclusive evidence for groundwater as the sole mechanism responsible for the formation of the seafloor terraces. However, we thought uncertainties were made clear in the paper and we have been careful to use words such as "potential", "propose", "likely" and included statements such as "more complex hydrogeological studies are required." (Row 2, page 2). The final sentence in the opening introductions reads:

"This study provides a framework for continued investigations involving in situ observations of potential groundwater seeping at selected terraces and semi-circular depressions along Baltic Sea coasts."

In any case, we believe it is important to clearly show uncertainties in science and are therefore happy to further emphasize them in our interpretations and also highlight in the discussion and abstract that we believe that the formation hypothesis should be tested with further observations.

In the revised version of the paper we have therefore done the following to meet the requirements of Referee #2 with respect to concern a):

1. Changed the title: "Potential links between Baltic Sea submarine terraces and groundwater seeping"

2. Changed the abstract to read "While submarine terraces can be produced by several processes, we interpret our results to be in support of the basic hypothesis of terrace formation initially proposed in the 1990s, i.e. groundwater flows through siltier permeable layers in glacial clay to discharge at the seafloor, leading to the formation of a sharp terrace when the clay layers above seepage zones are undermined enough to

collapse." This revision follows the suggestion of Virtasalo.

3. Changed the last sentence of the abstract to: "We propose that SGD through the sub-marine seafloor terraces is plausible and could be intermittent and linked to periods of higher groundwater levels, implying that to quantify the contribution of freshwater to the Baltic Sea through this potential mechanism, more complex hydrogeological studies are required."

4. Included more discussion on alternative formation mechanisms, specifically the topic raised by Referee #2 of liquefaction of course layers. We lead into this in the beginning of the discussion in the revised version by: "However, we cannot exclude that SGD not is the sole mechanism that can produce terraces in the seafloor similar to those we mapped in this study and, therefore, alternative formation mechanisms are discussed below."

We end the discussion with the following paragraph: "There are other mechanisms that potentially could have played a role in the formation of the seafloor terraces mapped in this study. For example, sliding and slumping of glacial varved clays has been suggested to occur due to liquefaction of layers during palaeoseismic events (Hutri and Kotilainen, 2007;Virtasalo et al., 2007). This could leave behind terraces at the seafloor formed in glacial clay. However, we do not observe any morphological evidence of sliding and most of the terraces we mapped occur in areas where the seafloor slopes at <1ïĆř and the terraces have nearly flat bases, as evident in the bathymetric profiles in Figures 2e, 3c and 4c. We also note that the terraces we mapped are widespread across the Baltic and systematically appear in glacial varved clay. It seems unlikely that slides would occur over such spatially large areas in several regions. Finally, the processes responsible for the formation of some of the terraces seems to be ongoing judging from the bottom photographs showing how small blocks of clay presently are falling down to form a sharp terrace (Fig. 6). While we cannot exclude that other processes formed the terraces mapped in this study, we interpret our results to be in support of the formation mechanism proposed by Söderberg and Flodén (1995). Our

study provides a geological and morphological framework for further research involving longer-term monitoring of potential SGD from the terraces."

With respect to concern b) we have thought of it. The fact is that we do not envision a process where ground water is vigorously escaping the glacial clay to form a seafloor terrace. Instead, it is more likely process where water is flowing slowly through permeable layers, much like is seen in gardens, where a very small hydraulic head is required. It should also be noted that the hydraulic head can be created from a far distance such as illustrated in Figure 9.

Virtasalo also brings up that the seeps that are illustrated in the paper are not connected to the terraces and questions why they are included. As we found an abundance of seeps in the multibeam water column data, the question whether they were related to the terraces or depressions in the seafloor immediately arouse since this could potentially be important for the interpretation. Furthermore, one of the main points with the paper is to document the geological context of the terraces, and here the occurrence of the seeps and their relation/no relation to the terraces is important. We have included the following sentence in the revised version on page 7 in order to better motivate why they are included:

"….Seeps from the seafloor were found to be a common feature (Fig. 2a), and the question immediately arose if the seeps were related to either terraces or depressions in the seafloor."

Detailed comments: Referee #2: "Page 2, line 25. Add "varved" between "glacial" and "clay". It would help the reader in case the varved clay structure was shortly explained already in this paragraph." Authors: Fixed

Referee #2: "Page 10, the first paragraph about glacial varved clays is very much dominated by Swedish publications. Please consider adding classical works by e.g. Sauramo in the discussion."

Authors: One of the classic works from Finnish geologist Sauramo on glacial clay is now also included with a reference. Page 10 in the revised version: "The use of glacial varved clay as a record documenting the ice retreat was adopted early also on the Finnish side of the Baltic basin (Sauramo, 1926). From the knowledge gained from these studies follows that. . .."

Referee #2: "Page 10, line 31. There is no compelling evidence for brackish Yoldia Sea northward and eastward from the south-central Sweden (Schoning 2001, Boreas). Yodia Sea is not necessarily relevant to the topic of the manuscript and it could be excluded from the discussion."

Authors: We believe that the sentence on the Yoldia Sea provides a geological time context, which by Referee #1 brought up as important. Since we prefer to keep this, we have included that a brackish Yoldia may have been constrained to the eastern part of the Baltic with a reference to Schoning, 2001.

"A brackish water phase called the Yoldia Sea, perhaps constrained to the central Swedish side of the Baltic (Schoning, 2001), followed the Baltic Ice Lake (Björck, 1995), however it would take several hundred years for the Baltic to become brackish after the drainage and deposition of varved glacial clay continued close to the retreating ice margin (Andrén et al., 2011)."

Referee #2: "Page 11, line 4, "irrefutably"."

Authors: Fixed

Page 11, lines 5-12. Perhaps the concretions formed already during glaciolacustrine or post-glacial lacustrine environments, and comparisons to present brackish-water Baltic Sea are not relevant?

Authors: Since there is no way to tell this, we have kept the comparison

Referee #2: "Page 12, lines 1-5. How about the O isotope composition of the post-glacial lacustrine phase? Perhaps it was similar to large lakes in Sweden and Finland

today?"

Authors: We can speculate that since d18O values have a relation to temp (and latitude) it is likely that they will be in similar range as the Baltic Sea today, however, the different Baltic Sea stages will matter as well. The larger lakes that we have in Sweden/Finland today are a little bit more negative in d18O values than the Baltic Sea. The Baltic Sea exhibits a gradient south to north but at the same time having a little bit higher values than the terrestrial freshwater lakes. This is caused by mixing in Atlantic Ocean/North Sea water and therefore the different stages of the Baltic sea probably exhibited the same type of changes when connected to the North Sea. When not connected, the Baltic Sea d18O values would have been more negative and during ice melt much more negative. We do not think it is helpful to the paper to include this in the paper, it is a bit beyond the scope and there are many uncertainties around it and no data records to reference.

---

## Author Response (AR2)

Date 2019-11-20

Dear Editor Susan Conway,
We are grateful for the acceptance of our manuscript after minor revision. We have inferred all the points you listed in a revised version of the manuscript now uploaded.

Regarding the point suggesting that we inform the readers about how the seeps were identified in the mid-water acoustic data, we noticed that we had called the approach we used in the methods for "flare identification", which of course caused confusion. This is now straightened out so the readers should easily be able to follow how we did to identify the seeps (sometimes called seep flares).

Sincerely

Martin Jakobsson

**Department of Geological Sciences**

| Stockholm University | Visiting address: | Phone: +46-8-164719 |
| --- | --- | --- |
| | Svante Arrhenius Väg 8 | Telefax: +46-8-6747897 |
| | 106 91 Stockholm, Sweden | Email: martin.jakobsson@geo.su.se |